# Hierarchical Analysis of Factors Determining the Impact of Forest Certification in Mexico

Emily García-Montiel [1], Frederick Cubbage [2], Alberto Rojo-Alboreca [3], Miriam Mirelle Morones-Esquivel [1], Concepción Lujan-Álvarez [4], Eusebio Montiel-Antuna [1], Pablito Marcelo López-Serrano [5], Fernando Pérez-Rodríguez [6] and José Javier Corral-Rivas [1,*]

1 Facultad de Ciencias Forestales, Universidad Juárez del Estado de Durango, Río Papaloapan, Valle del Sur, Durango 34120, Mexico
2 Department of Forestry and Environmental Resources, North Carolina State University, Raleigh, NC 27695, USA
3 Escola Politécnica Superior, Universidad de Santiago de Compostela, Campus Universitario S/n, 27002 Lugo, Spain
4 Facultad de Agrícolas y Forestales, Universidad Autónoma de Chihuahua, Km. 2.5 Carretera Delicias a Rosales, Campus Delicias, Cd. Delicias, Chihuahua 33000, Mexico
5 Instituto de Silvicultura e Industria de la Madera, Universidad Juárez del Estado de Durango, Boulevard Guadiana 501, Fraccionamiento Ciudad Universitaria, Durango 34120, Mexico
6 Föra Forest Technologies sll, Campus Duques de Soria s/n, 42004 Soria, Spain
* Correspondence: jcorral@ujed.mx; Tel.: +52-618-122-9423

**Abstract:** Forest certification is a private, voluntary and market-driven instrument designed to promote responsible forest management. This paper focused on the FSC and the NOM NMX-AA-143-SCFI-2008 schemes used in Mexico for the certification of sustainable forest management. In this paper we used the analytical hierarchical process (AHP) to study the factors that determine the main impacts of forest certification in México. A panel of 30 experts was selected as decision-makers to find which principles, criteria and indicators are considered as the most relevant while implementing forest certification. For decision-makers, the environmental principle occupied the first place with 40.26% of importance in the implementation of forest certification, followed by the social principle, and the economic principle with 32.15% and 27.59% of importance, respectively. Regarding the criteria, forest management and production, biodiversity, and forest protection were considered to be the most relevant. Regarding the indicators, the results indicated that forest certification in Mexico can have a positive impact on the existence of educational institutions, community services such as water, energy, medical services and drainage, the quality of the forest management plans, investment in forest management, machinery and equipment, environmental services, recreation, tourism, research, development and community education, planning for the conservation of biological diversity, and planning for biodiversity conservation.

**Keywords:** Analytical Hierarchy Process (AHP); principles; criteria; indicators

## 1. Introduction

The forest sector directly and indirectly impacts various industries and economic activities. In addition, forests provide various environmental services at local and global scales. The demand and conservation of these services has increased strongly in recent decades, which has demanded the creation of forest certification as a mechanism to promote "sustainable forest management" and to seek for the sustainability of forests [1]. In this sense, forest certification is a voluntary instrument inserted into the market with the purpose of encouraging change in the forestry sector, mainly in the certification of forestry operations and their companies in the value chain. It aims to link producers and consumers by awarding an eco-label to forest products that comply with a set of environmental and social criteria and indicators [2]. Furthermore, it is a non-binding instrument that seeks to

assess forest management, legality verification conditions, chain of custody, eco-labelling and trademarks to promote sustainable forest management, all with the aim of stimulating ethical trade and improving market access through economically viable, environmentally sound and socially beneficial management [3].

The idea of certification in forest management is simple. A timber enterprise (individual or communal) demonstrates that it operates to high standards in both the ecological and socio-economic aspects of forest management. Consequently, it gains approval from a third-party certifying agency and gains access to markets willing to pay higher prices for sustainably harvested forest products [4].

Since its inception more than two decades ago, forest certification has proliferated in developing countries and its standards have become increasingly important in sustainable forest management [5,6]. As a member of ISEAL (International Social and Environmental Accreditation and Labeling Alliance), with more than 200 million hectares certified, the Forest Stewardship Council (FSC) is one of the most recognized forest certification systems both in forest management and the chain of custody. It aims to assure consumers that the wood products they buy come from well-managed forests that respect specific environmental, social and economic principles and criteria [7]. As of June 2022, the FSC has certified 217,915,640 ha in 89 countries [8]. Mexico's FSC National Standard for Responsible Forest Management (ENMFR) was adapted to the national scale to reflect the diverse legal, social and geographical conditions of forests and plantations in Mexico. The standard can be used by all types of organizations working in natural forests and plantations in Mexico, including small and low-intensity managed forests (SLIMF) and it is also applicable to a large group of non-timber forest products, such as candelilla wax, nopal, pine resin, bamboo and others [8].

In Mexico there are 1,374,726 ha certified under FSC [8]. This forest certification was pioneered in the mid-1990s by two non-governmental organizations: the Mexican Civil Council for Sustainable Silviculture (in Spanish, Consejo Civil Mexicano para la Silvicultura Sostenible, CCMS), which has focused on community forestry, and the Rainforest Alliance's SmartWood programme, which was an FSC certifier in Mexico [9], and which now has its own sustainable agriculture certification standard for producers and supply chains [10]. Years later FSC, ejidos, communities, and government institutions such as CONAFOR, SEMARNAT and PROFEPA felt there was a need for a simple Mexican forest certification program. They cooperated to develop a Mexican Forest Certification System. The Mexican Council of Forest Certification was established in September 2008, and is composed of national organizations of forest producers and industrial chambers grouped in the forest business council aiming to promote sustainable forest management and consumption of forest products from legal and certified sources.

The Mexican certification system was created as a unified strategy that incorporates the existing forest certification instruments in a single institutional policy to promote sustainable forest management in Mexico. It was promoted by the institutions responsible for the national forest development (CONAFOR and SEMARNAT).

The generic standard of the Mexican Official Standard: NMX-AA-143-SCFI-2008 has a twofold objective: (i) technical: to assess the forest management, ensuring compliance with the economic, social functions and ecological forestry; and (ii) commercial: improved market access and distribution of products from certified forests. The system was developed by the Mexican Accreditation Entity (EMA), which in turn authorizes the Association for Standardization and Certification (ANCE) as the certification body [9].

According to Auld et al. [11], forest certification has contributed to the use of better forest management practices at the forest management unit (FMU) level. However, Gullison [12] reported that forest certification at the landscape level has not reduced pressure on forests, and therefore has not reduced deforestation rates in tropical countries. On the other hand, certified companies face higher operating costs necessary to maintain certification and compliance with scheme requirements, which are only partially offset, not by an increase in the selling price of certified products, but by the company's ability to establish

new business relationships with a consequent increase in sales [13]. Thus, more research is needed on how certification is implemented and on how it promotes a sustainable forestry and adaptive forest management [14]. Indeed, little is known about "how" certification bodies, auditors' rules, and audit procedures shape the implementation of the standards, even though what they define as non-compliance guides the company in reforming their forest management through the required corrective action [15].

Many forest stakeholders now agree on the need to perform formal evaluations of the empirical environmental, social, economic and policy impacts of certification in natural forests in developing countries. This requires knowledge of how certification is applied and an examination of the indicators of country-specific standards, rules and practices of certification bodies and their auditors [13,16]. This lack of knowledge about field surveys may be related to the fact that it is costly, labor-intensive, and time-consuming to carry out surveys that generate such information. In Mexico there were several FMOs (Forest Management Organizations) that were ejidos and communities with experience in sustainable forest management when the certification movement started [4]. The ejido in Mexico is one of the land tenure modalities that make up agrarian social property, the only difference between ejido and community is the legal form in which they are constituted. Ejidos and communities are rural communities that manage their forests with some level of government control [17]. In this context, "government control" means that they must practice forest management according to federal laws, mainly subject to the Norma Oficial Mexicana "NOM-152-SEMARNAT-2021", which specifies the guidelines and particular requirements for management plans related to the use of timber resources in coniferous and tropical forests and in arid regions of Mexico [18]. Forestry production in Mexico is subdivided into two main categories: timber and non-timber. The former is made up of woody materials. The latter is made up of resins, fibers, gums, waxes, leaves, stalks, stems, seeds, rhizomes, and forest soil, among others. In 2018, according to SEMARNAT's forest statistical yearbook, there were 13,971 timber harvesting authorizations, corresponding to an area of 6,078,986 hectares distributed to communities, ejidos and private landowners who are the main participants in the forest industry supply chain. The main products in Mexico are squaring, veneer and plywood, firewood, poles, piles and charcoal [19]. Timber extraction is divided into cutting and chopping, hauling, loading, freighting and cleaning. In Mexico, the forestry industry is still based on semi-automated technology and most of the companies in Mexico are raw material processors and not traders.

To analyze the impact of forest certification, it is necessary to identify and prioritize from a large number of principles, criteria and indicators regarding their preferences for rational priority setting. The selected principles, criteria and indicators may be different among countries and show substantial and important impacts on decision making about forest certification. The multi-criteria decision-making method (MCDM) can be used to systematically identify preferences for principles, criteria and indicators involved in forest certification. Specifically, the analytical hierarchical process (AHP), developed by Saaty [20,21] has been applied in many different research fields, including environmental, economic and social issues e.g., [22–26], and in other applications. It is also used to establish a ranking of indicators for sustainability [27]. AHP has also been used to assess experts' perceptions of the level of difficulty in implementing forest certification [28], and to assess how key stakeholders relate certification to sustainability, and its implications for sustainable forest management [29].

The AHP is a technique used for multi-criteria decision making by pairwise comparison. With the help of a scale, this set of pairwise comparisons allows for the estimation of priorities (weights) among criteria (and among alternatives, if any) and, finally, a hierarchy according to their importance in the final decision [21]. The AHP process facilitates a more complete understanding of the importance of each of the aspects that are evaluated in forest certification processes, as well as improving skills in predicting and planning the implementation of a forest certification scheme. It also helps to prepare, and to cope with, audits of forest certification, prioritizing the activities that need to be given more attention

in order to achieve and maintain the certification. AHP provides a framework for selecting a preferred alternative from a set of potential solutions to a problem as well as planning and management decisions [30]. Based on the above, the objective of this paper was to identify the principles, criteria and indicators of more impact on FSC forest certification in Mexico. We used pairwise comparisons of the AHP method made by a group of experts to the principles, criteria, and indicators involved in Mexico's FSC National Standard for Responsible Forest Management and in the NOM NMX-AA-143-SCFI-2008.

## 2. Materials and Methods

### 2.1. Analytical Hierarchy Process (AHP)

This study used the AHP process introduced by Saaty [20], which is effective in dealing with complex decision making, and can help the decision maker to prioritize and make the best decision. By reducing complex decisions to a series of pairwise comparisons and then synthesizing the results, AHP helps to capture the subjective and objective aspects of a decision. In addition, AHP incorporates a useful technique for checking the consistency of the decision-maker's evaluations, thus reducing bias in the decision-making process. The methodology used to perform this analysis in this study is described below:

1. Establishment of the decision tree or outline: the principles, criteria and indicators that have been selected and how they have been structured in 3 levels (level 1: principles, level 2: criteria, level 3: indicators) (see Table 1);

**Table 1.** Structure of the decision framework used in this study.

| Principles | Criteria | Indicators | Verifiers |
|---|---|---|---|
| Economic Principle | 5 | 18 | 30 |
| Environmental Principle | 2 | 4 | 16 |
| Social Principle | 8 | 10 | 12 |

2. Selection of a panel of experts: for this study, a panel of 30 experts was selected as decision-makers based on the relevance of the principles, criteria and indicators considered in the assessment tool. This group consisted of five professors and researchers of forest management and forest certification, five members of government agencies with functions related to forest certification, ten members of non-profit associations promoting certification, and ten managers of forest certified properties. All decision-makers who participated in the study were selected directly because of their good knowledge and expertise of the forest certification process in Mexico;

3. Paired comparison: each of the decision-makers made their decisions separately and were able to make up to three replications to reduce the level of inconsistency;

4. Perform the calculations of: the maximum eigenvalue, the CI consistency index, the CR consistency ratio and the normalized values for each principle, criterion, indicator and verifier, using hierarchies, which then cast the mental model in the structured prototype. At the first level, the principles are the main focus. At the next levels, criteria, indicators are evaluated. The AHP uses the paired comparisons to incorporate the preferences of the experts or decision-makers among the elements $a_{ij}$ of the model; using the fundamental scale 1, 3, 5, 7 and 9, to integrate the judgements or evaluations of the decision-makers (i.e., the value of $a_{ij}$ = 1 if $i$ and $j$ are equally important, 3 if $i$ is slightly more important than $j$, 5 if $i$ is more important than $j$, 7 if $i$ is strongly more important than $j$, 9 if $i$ is absolutely more important than $j$). The relationship is strictly positive to eliminate the ambiguities of zero and infinity; using the right eigenvector method to obtain the local priorities, the hierarchical composition principle to calculate the global priorities, and a multidivisional linear form to estimate the total priorities. This allows for the resolution process to assess the consistency of the decision-maker in making judgements, and, finally, the priorities are given on a ratio scale [31].

A binary comparison of a criterion $i$ with a criterion $j$, is represented as $a_{ij}$, and is shown as follows:

$$a_{ij} = 1/a_{ji} \tag{1}$$

The process of paired comparisons with $n$ elements at one level of the hierarchy generates a matrix equation:

$$[A][a] = n[a] \tag{2}$$

where: [A] is a square matrix containing the set of paired comparisons $a_{ij}$ and its reciprocal with $a_{ji} = 1/a_{ij}$, $n$ is the dominant Eigenvalue of [A], and [a] is the associated Eigenvector representing the weights of the criteria or alternatives. For a level in the hierarchy with $n$ criteria, the number of comparisons to be performed are:

$$\text{No. of comparisons} = (n(n-1))/2 \tag{3}$$

For an ideal decision-maker, it follows that:

$$a_{ij} = a_i/a_j \tag{4}$$

The actual situation of the paired comparison, in which there are no consistencies, generates the following equation:

$$[R][a] = \lambda\max[a] \tag{5}$$

where: [R] is called a perturbation of [A], $\lambda$max is the dominant Eigen value in the $+\,\Re$, and [a] is its eigenvector. The inconsistency coefficient is determined as:

$$CI = (\lambda\max - n)/(n-1) \tag{6}$$

In the evaluation process, the inconsistency ratio is required to be less than 10% to accept the calculation of the disturbance matrix. The inconsistency ratio is calculated as:

$$RI = CI/CIA \tag{7}$$

where CIA is a random inconsistency index.

*2.2. Decision Tree*

The principles and criteria used in this work are part of Mexico's FSC National Standard for Responsible Forest Management (SFM-STD) and of the NOM NMX-AA-143-SCFI-2008 used in Mexico for the certification of sustainable forest management. These standards specify the minimum requirements for certifying a Forest Management Organization in Mexico (FMO), and hierarchically organized into principles, criteria, indicators and verifiers or guidance notes. The principles and criteria describe the essential elements or rules of forest certification (general and intangible objectives), they were first established by FSC to be common and applicable to all forests worldwide. Each principle is supported by several criteria that provide a way of judging whether the principle has been met in practice, and then each criterion is supported by several indicators that provide a way of judging whether the criteria have been met in practice. The indicators are practical and specific standards that a FMO must meet. They were developed and adapted to the forests of Mexico and are measured by the auditors to decide on the certification of each FMO. Thus, the hierarchical structure of the SFM-STD makes it possible to transform intangible objectives (principles and criteria) into measurable elements (indicators). Finally, the verification or guidance notes indicate where auditors can look for information on compliance indicators. For the present work, a decision tree was established, divided into four levels or groups of elements, based on the hierarchical process of the SFM-STD, and on the experience of the authors of forest certification in Mexico (Table 1).

The cluster or level 1 of the decision-making scheme (root cluster) included only three elements (called principles, P), corresponding to the economic, environmental and social aspects implicit in the ten principles of the FSC certification scheme, as well as the nine principles established by NOM NMX-AA-143-SCFI-2008 used in Mexico for the certification of sustainable forest management.

Each principle (level 1) was subdivided into three further levels, which for the purposes of this paper were successively referred to as criteria (level 2), indicators (level 3) and verifiers (level 4). The economic principle includes 5 criteria, 18 indicators and 30 verifiers. The environmental principle includes 2 criteria, 4 indicators and 16 verifiers. The social principle includes 8 criteria, 10 indicators and 12 verifiers. In total, 108 elements were selected for the decision framework: 3 principles (level 1), 15 criteria (level 2), 32 indicators (level 3) and 58 verifiers (level 4).

The decision-making scheme discussed above was introduced in the VSElephas 1.0® software for collaborative decisions on Cloud [32], the 30 members of the expert group (researchers, forest public workers, NGO members and forest managers) were invited to make their decisions, by means of an e-mail with a password to access the software, a description of the work to be done with the program, and a deadline for carrying out the judgments. Each decision-maker was requested to carry out three repetitions for each decision.

### 2.3. Ranking of Options

Once all the judgements had been recorded by the decision-makers, the estimated importance of the principles, criteria, indicators and verifiers was calculated using the VSElephas® software. In this case, as it was only a matter of establishing a hierarchy of elements (or criteria) and not of comparing alternatives for each criterion, the calculation was reduced to establishing unit matrices formed by the values of the pairwise comparisons of the elements, which were then weighted by the higher-level criteria from which they emanated. To minimize the combinatorics of the AHP methodology (i.e., to reduce the number of pairs to be evaluated), clustering was performed, as well as to establish the profile of the decision-makers to define the importance rating between the combined pairs.

The matrix of option scores contained an indicator of the validity or consistency of the decisions made, called the degree of inconsistency, which was obtained by comparing the unitary matrix with a homogeneous matrix [20,21]. According to Saaty [21], if this indicator is higher than 0.1 (or higher than 10%) the decision can be considered inconsistent, and should be repeated, although some authors (e.g., [33]) report that such inconsistency is meaningless. After each evaluation of a cluster, the VSElephas® program shows the user the value of that inconsistency, also marking it on a colour scale (green: null or low, yellow: permissible, and red: unacceptable).

However, by allowing up to three repetitions of the same grouping decision, it can happen, that the first evaluation is inconsistent and that in successive repetitions the inconsistency is minimized. Finally, the calculations concludes with the synthesis process, which results in the net weight of each of the elements that make up the decision tree, which can be hierarchized for each of the established levels: principles, criteria, indicators and verifiers.

### 3. Results

Tables 2–4 show the ranking of principles, criteria and indicators (results for verifiers are not shown because they are indeed a way of judging whether the indicators have been met in practice). As can be seen in Table 2, for decision-makers or experts, the environmental principle occupies the first place with 40.26% of importance in the implementation of forest certification in Mexico, followed by the social principle, and the economic principle with 32.15% and 27.59% of importance, respectively. This means that for forest certification experts in Mexico the environmental principle is considered the most important, compared to the social and economic principles.

**Table 2.** Importance ranking in percentage of forest certification principles assessed by the Analytical Hierarchy Process in this study.

| No. | Principle | Percentage (%) |
|---|---|---|
| 1 | Environmental principle | 40.26 |
| 2 | Social principle | 32.15 |
| 3 | Economic principle | 27.59 |

In terms of the 15 forest certification criteria (Table 3), the two criteria of forest management and production, and biodiversity and forest protection have been weighted at 21.62% and 18.64%, respectively, showing that decision-makers consider them the most important criteria,. Economically oriented criteria, including investments; production and marketing; cost and benefit distribution did rank in a second grouping of importance, from 7.57% to 6.45%. Educational and community-oriented criteria fell in the third ranking category from 5.69% to 4.49%. Forest security concerns, retaining viable local populations, and ethnic composition fell perhaps into a fourth level, ranging from 3.81% to 3.57%. The lowest grouping of criteria were spiritual and cultural values, capital stocks, and aesthetic and recreational values (2.61% to 2.16%).

**Table 3.** Importance ranking in percentage of forest certification criteria assessed by the Analytic Hierarchy Process in this study.

| No. | Criteria | Code | Percentage (%) |
|---|---|---|---|
| 1 | Forest management and production | MPF | 21.62 |
| 2 | Biodiversity and forest protection | BPB | 18.64 |
| 3 | Investment in forestry, tourism and recreation | ISFTR | 7.57 |
| 4 | Production, consumption and marketing of goods and services | PCCBS | 7.17 |
| 5 | Distribution of costs and benefits to the community | DCBC | 6.45 |
| 6 | Educational values: Institutions for Human Resource Training | VEIFRH | 5.69 |
| 7 | Community Services: Public Services | SC | 5.38 |
| 8 | Community participation | PPC | 4.53 |
| 9 | Community autonomy, land tenure and coexistence: rights and responsibilities of tenure and use. | ACDR | 4.49 |
| 10 | Losses due to clandestine/robbery: forest damage due to clandestine and robbery of flora and fauna during the last year. | PCRA | 3.81 |
| 11 | Emigration/immigration of the population: need of rooting of the population | EINA | 3.72 |
| 12 | Presence of ethnic groups: minority ethnic groups in the community | GPED | 3.57 |
| 13 | Spiritual and cultural values | VESP | 2.61 |
| 14 | Capital stocks | EXCP | 2.59 |
| 15 | Aesthetic and recreational values | VER | 2.16 |

There were about twice as many indicators (31) to be ranked, which probably decreased the range in AHP percentage scores—from a high of 5.69% for educational institutions to a low of 1.02% for wood harvest characteristics. The results of the AHP process carried out in Mexico at the indicator level (Table 4) seemed to be less consistently grouped into broad subject matter themes than the criteria were. Five diverse indicators ranked the highest: the existence of educational institutions (relative weight of 5.69%), community services such as water, energy, medical service and drainage (relative weight of 5. 38%); the quality of the forest management plans (relative weight of 4.96%); investment in forest management, machinery and equipment; environmental services, recreation, tourism, research, development and community education (relative weight of 4.88%); and planning for biodiversity conservation (relative weight of 4.35%).

Results of the AHP ranking of the rest of the forest indicators in Mexico varied considerably. The next rankings covered monitoring (3.94%), production diversification, community ownership, development regulations, and best management practices (3.62%). The following indicators can be classed in order of addressing approximate areas of sociological

(3.57%), ecological, management, sociological, ecological, and economic subjects (1.02%). This suggests there is less of a trend in the ranking importance of any broad grouping of importance at the indicator level compared to the criteria level.

**Table 4.** Importance ranking in percentage of forest certification indicators assessed by the Analytical Hierarchy Process in this study.

| No. | Indicators | Code | Percentage (%) |
|-----|------------|------|----------------|
| 1 | Existence of educational institutions | IED | 5.69 |
| 2 | The community has services such as water, energy, medical service and drainage supported by forestry. | SAE | 5.38 |
| 3 | Forest management program | PMF | 4.96 |
| 4 | Investment in forest management, machinery and equipment, environmental services, recreation, tourism, others: Investment for research, development and community education | IMM | 4.88 |
| 5 | Planning for the conservation of biological diversity | PCBV | 4.35 |
| 6 | Implementation/Efficiency of management monitoring | IMJ | 3.94 |
| 7 | Diversification of production | DPN | 3.72 |
| 8 | There are strategies for strengthening community ownership and its level of implementation | ACNP | 3.72 |
| 9 | Existence and application of regulations for the development of the community | EARD | 3.72 |
| 10 | Use and monitoring of best management practices | UMPM | 3.62 |
| 11 | There are agreements with indigenous peoples who have customary/traditional rights in the property and are taken into account in the decision making on the property | CPIN | 3.57 |
| 12 | Forests or attributes of high conservation value | BAVC | 3.30 |
| 13 | Product Market: there are market studies and their operation (application) for the commercialization of timber products | MP | 3.17 |
| 14 | Sustained yield/crop allowed/adjacency constraints | RSCP | 3.09 |
| 15 | Forest inventory for the purpose of elaborating the management plan | IFFPM | 2.29 |
| 16 | Meet "Green-up" standards (habitat connectivity) | GUP | 2.16 |
| 17 | Education on natural/forest resources: the population participates in reforestation, garbage collection, protection and environmental education | ERN | 2.05 |
| 18 | Area and percentage of forest land affected by illegal logging | STI | 1.91 |
| 19 | Presence of illegal hunting | CCI | 1.90 |
| 20 | Participation in forest decision-making: assemblies are held where the community is informed about the forest use made on their land and the benefits generated for it | PPF | 1.84 |
| 21 | Protection of threatened species | PEA | 1.74 |
| 22 | Decrease forest conversions | RXB | 1.69 |
| 23 | Protection of riparian ecosystems or water sources | PERC | 1.64 |
| 24 | Have costs been estimated to maintain certification? | CMC | 1.58 |
| 25 | Forest health protection | PSB | 1.34 |
| 26 | Total cost to the community for production, forestry and other activities: Total cost of production | CTCP | 1.31 |
| 27 | Decision on profits and creditors: type of mechanisms to share profits | DSGA | 1.30 |
| 28 | There is an action plan to find customers in international market | PACMI | 1.30 |
| 29 | Cleaning of cutting areas | LAC | 1.16 |
| 30 | Who has borne most of the certification costs? | CCN | 1.10 |
| 31 | Annual volume of harvested wood, quantities and type of products | VAMC | 1.02 |

## 4. Discussion

This paper presented an analysis that identified the importance of the principles, criteria and indicators of FSC forest certification in Mexico through the application of the AHP process to a group of experts. Based on the criteria level of certification standards, the experts who participated in the AHP analysis ranked the environmental principles as the most important and positive effects of forest certification in Mexico, followed by the social and then economic principles. These results are consistent with the findings of Tamarit-Urias [34], who mentioned that forest certification in Mexico has generated environmental benefits but has not yet contributed significantly to the socio-economic development of Mexico's forest regions. Rodriguez [35] also reported that forest certification in Mexico

has mainly brought changes in the environmental and silvicultural aspects, specifically a change in the attitudes of foresters, showing greater conviction for the good care and management of their forests, and above all, the existence of well-managed and conserved forests. This coincides with the hierarchical order of the criteria showed in this study, where environmental indicators are positioned in the first places.

These criteria, ranking environmental factors highest at the principle level, carried over to the criteria level, which ranked the related forest management and production and biodiversity and forest protection criteria the highest. However, at the level of certification criteria, the economic components of investment, diversification of production, tourism or recreational activities, as well as production, consumption and marketing of goods and services were positioned in the second grouping of importance. This makes sense if these results are perceived as criteria that need special attention and a marketing strategy beyond simple certification. These results are consistent with the findings of others [13,36,37] who recommend that, in addition to the productive diversification suggested by FSC forest certification, investment and an optimal market strategy are decisive in order to obtain the benefits of the competitive advantage of companies that market certified products. The ranks for several social and community criteria do follow immediately after the economic criteria, so the difference in the order from the principles category is not large.

In terms of the ranking of indicators, results indicate that the top ten places of importance, according to the experts, are occupied by indicators that evaluate the social and environmental benefits that forest certification leaves for the communities, which are related specifically to the provision of basic services that improve the quality of life of the inhabitants, and to the incorporation of new forestry practices for biodiversity conservation such as the implementation of monitoring systems for flora and fauna. These results are in agreement and supported by other studies [38–43] that reported that forest certification in Mexico and in other countries has contributed to increasing social infrastructures and to improving the quality of the technical service providers. Broad reviews by FAO [44] and the World Bank [45] identified that the basic and important indicators of forest certification were the conservation of biological diversity and ecological functions; measures to maintain or enhance the multiple environmental benefits provided by forests; prevention or minimization of the environmental impacts of forest use; effective forest management planning; active monitoring and evaluation of relevant areas of forest management; and maintenance of critical forest areas and other critical natural habitats affected by forest operations.

These indicators coincide with the results obtained in this study in which experts prioritize environmental and social aspects as the most relevant to meet the requirements of forest certification in comparison to the economic benefits which are still lacking in Mexico, where nearly all certified ejidos and communities sell their timber at the same price as non-certified timber. In addition, there are simply fewer economic indicators that exist in the Mexican forest certification system, so they are apt have a lower proportion of high ranks regardless. The Mexican system is also patterned closely after the FSC system, which also places more focus on environmental and social aspects.

On the other hand, forest owners in some European countries recognize that certification leads to changes in other aspects besides environmental ones, in a deeper sense, i.e., in Slovakia owners see certification as an important tool for the promotion of the sustainable use of forest resources, improving SFM practices, and assisting them to penetrate new markets; they identified more with the fact that certification helps to improve market access and increase profit margins, and this was due to the demand for FSC-certified wood driven by wood processing companies operating within the chain supplying multinational furniture industry plants based in the country. This motivates forest owners to gain certification and thus new customers [46]. The same study mentioned that certification is also seen as a tool for promoting learning and facilitating the exchange of experiences to ensure sustainable forest management and to ensure the achievement of forestry-related policy objectives and improve their functioning, intangible but valuable outcomes and benefits. In Italy, the development of the Italian Sustainable Forest Management certification scheme since 2001

has shown how important non-wood forest products are for rural and forest communities, in some areas much more than timber production itself [47], which is a broader vision of the diversification of production.

## 5. Conclusions

The results of this study, that used an AHP analysis of the impact of the Mexico forest certification system, are interesting and nuanced. Based on the AHP rankings by the panel of experts, the most important subject areas and factors differ somewhat, depending on whether one considers their importance at the principle, criteria, or indicator level. These differences might reflect (1) an absolute importance of different factors, but they also may reflect both (2) some inconsistency in the alignment in the consistency of the environmental, social, and economic components at the three certification levels, and (3) some ambiguity in the wording of the criteria and indicators, which do combine a lot of quite diverse words and factors in each stated principle or indicator.

Furthermore, it is important to note that numerical rankings should not be taken as dismissing the importance of even the lower ranked indicators in some respects, and in different ejidos or forests. An immense amount of effort has gone into developing, rationalizing, and improving the Mexican system of forest certification, and the principles, criteria, and indicators represent a thoughtful balance of environmental, social, and economic components. The AHP system must select higher or lower ranks by definition, but that does not mean that lower ranks are irrelevant or do not need to be implemented and audited. Lower ranks may provide factors that can be considered for improvement or exclusion of subsequent continuous improvement of the certification system, but those indicators still must absolutely be followed to receive certification under the current Mexican system. This approach is indeed the case for all forest certification systems.

We can conclude that the environmental and the social principles were the most prominent in the opinion of the experts, and perhaps bear commensurately greater attention by forest owners seeking forest certification. However, at the secondary principle level, the economic principles ranked slightly higher than the social principles, indicating perhaps that they are the "sine qua non."—the essential condition; without which there is none. Indeed, there may be fewer economic indicators in forest certification because markets and trade essentially drive many of the forest management and business decisions, and need less standards to influence those components of sustainable forest management. Overall, these results can help forest owners and certification system administrators identify and understand the perceptions of experts about the relative importance of the diverse components of Mexican forest certification and its implementation, and which areas that should receive more attention along the forest management chain and the administration of ejidos, communities, and private properties.

**Author Contributions:** E.G.-M., C.L.-Á., M.M.M.-E. and E.M.-A. conceived, designed and performed the experiments and wrote the manuscript. F.C., A.R.-A., F.P.-R., J.J.C.-R. and P.M.L.-S. analyzed the data and revised the manuscript. All authors have read and agreed to the published version of the manuscript.

**Funding:** This research was funded by El Consejo Nacional de Ciencia y Tecnología (CONACYT), grant number (362184).

**Data Availability Statement:** Not applicable.

**Acknowledgments:** The first author expresses thanks to CONACYT for her doctoral awarded scholarship (Grant 362184) and to the experts who contributed their time to this study.

**Conflicts of Interest:** The authors declare no conflict of interest.

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
