# Peer review of "Hierarchical Analysis of Factors Determining the Impact of Forest Certification in Mexico"

_forests, doi:10.3390/f13122093_

Round 1
Reviewer 1 Report
The paper present interesting results related to certification schemes in Mexico, particularrly what principle/criteria/indicators has the highest impact on FSC implementation on national level. I agree with methods used - as panel discussion and AHP methods are very useful when approaching the stakeholders and the aim is to observe their experiences/opinions. I have only small recommendations:
r. 42 – „good forest management“– consider replacing it with better wording, maybe „preferential forest management practices“ / „sustainable forest management“ or something similar used related to forest certification. Even in the ENMFR mentioned in row 66 is responsible forest management. I do not understand the term „good“
r. 64-65 : „Currently, the FSC has certified xxx ha in xx countries“ – even in citations the month when FSC site was accessed is present, authors should slightly change the beginning of the sentence: f.e. “As of Jun 2022, the FSC has certified…” – or similar.
r. 137 – add the author name : “….AHP process introduced by Saaty [19],…”.
r. 236/table 1 – the number of criteria from the table seems to be 13, but, in the text, 15 criteria are mentioned. Why?
The discussion section - the results are well-discussed on the national level, maybe a short paragraph about similar studies around the world should be added. For example, many Europan studies exist on the importance of certification schemes and why forest owners engage in these schemes.
Author Response
Thank you for your comments, please see attachment.

Reviewer 2 Report
1. It is good if the author can describe in general the Forest Management System in the country with regards to government regulations if any and enforcement agencies etc. Is the jurisdiction under state or federal government?
2. Supply chain of forest product from logging to saw mill to manufacturing to distribution to give a clear view to the readers.
3. Page 2 line 99 - What is ejidos?
4. Line 272 – not shown
5. Page 11 Line 452 – no reference list?
Author Response

(The authors gave the same response as above.)
